# The muscle proteome reflects changes in mitochondrial function, cellular stress and proteolysis after 14 days of unilateral lower limb immobilization in active young men

Thomas M. Doering[1,2]*, Jamie-Lee M. Thompson[2], Boris P. Budiono[3], Kristen L. MacKenzie-Shalders[2], Thiri Zaw[4], Kevin J. Ashton[2], Vernon G. Coffey[2]*

1 School of Health, Medical and Applied Sciences, Central Queensland University, Rockhampton, Queensland, Australia, 2 Bond Institute of Health and Sport, Faculty of Health Sciences and Medicine, Bond University, Gold Coast, Queensland, Australia, 3 School of Dentistry and Medical Sciences, Charles Sturt University, Port Macquarie, New South Wales, Australia, 4 Australian Proteome Analysis Facility, Macquarie University, Macquarie Park, New South Wales, Australia

* t.doering@cqu.edu.au (TMD); vcoffey@bond.edu.au (VGC)

**Data Availability Statement:** Proteomics data have been deposited to the ProteomeXchange

## Abstract

Skeletal muscle unloading due to joint immobilization induces muscle atrophy, which has primarily been attributed to reductions in protein synthesis in humans. However, no study has evaluated the skeletal muscle proteome response to limb immobilization using SWATH proteomic methods. This study characterized the shifts in individual muscle protein abundance and corresponding gene sets after 3 and 14 d of unilateral lower limb immobilization in otherwise healthy young men. Eighteen male participants (25.4 ±5.5 y, 81.2 ±11.6 kg) underwent 14 d of unilateral knee-brace immobilization with dietary provision and following four-weeks of training to standardise acute training history. Participant phenotype was characterized before and after 14 days of immobilization, and muscle biopsies were obtained from the *vastus lateralis* at baseline (pre-immobilization) and at 3 and 14 d of immobilization for analysis by SWATH-MS and subsequent gene-set enrichment analysis (GSEA). Immobilization reduced vastus group cross sectional area (-9.6 ±4.6%, P <0.0001), immobilized leg lean mass (-3.3 ±3.9%, P = 0.002), unilateral 3-repetition maximum leg press (-15.6 ±9.2%, P <0.0001), and maximal oxygen uptake (-2.9 ±5.2%, P = 0.044). SWATH analyses consistently identified 2281 proteins. Compared to baseline, two and 99 proteins were differentially expressed (FDR <0.05) after 3 and 14 d of immobilization, respectively. After 14 d of immobilization, 322 biological processes were different to baseline (FDR <0.05, P <0.001). Most (77%) biological processes were positively enriched and characterized by cellular stress, targeted proteolysis, and protein-DNA complex modifications. In contrast, mitochondrial organization and energy metabolism were negatively enriched processes. This study is the first to use data independent proteomics and GSEA to show that unilateral lower limb immobilization evokes mitochondrial dysfunction, cellular stress, and proteolysis. Through GSEA and network mapping, we identify 27 hub proteins as potential protein/gene candidates for further exploration.

Consortium via the PRIDE partner repository with the dataset identifier PXD034908 (http://proteomecentral.proteomexchange.org/cgi/GetDataset?ID=PXD034908).

**Funding:** This study was funded by the Collaborative Research Network for Advancing Exercise and Sport Science (CRN-AESS - 201202) scheme awarded to VGC and KJA by the Department of Education and Training Australia. The funding body played no role in the study design, data collection and analysis, decision to publish, or preparation of the manuscript.

**Competing interests:** The authors have declared that no competing interests exist.

## Introduction

Skeletal muscle unloading due to immobilization results in substantial muscle remodelling and atrophy in response to the reduced mechanical loading and contractile activity. These consequences of immobilization expose otherwise healthy individuals to deleterious locomotive and metabolic consequences [1]. Indeed, periods of muscle unloading due to knee-brace or cast immobilization result in significant reductions in the cross-sectional area (CSA) and strength of muscles acting on/across the knee joint. However, variable phenotypic data have been reported despite similar immobilization protocols across a 14 d period. For example, declines in muscle CSA have been reported to range from 5.0 ± 1.2% to 8.4 ± 2.8% [2–5], with associated strength losses of ~6–25% [2, 3, 5].

In response to skeletal muscle unloading, muscle protein equilibrium shifts towards a net loss of proteins. Whether reductions in muscle protein synthesis (MPS) or elevations in muscle protein breakdown (MPB) are the primary drivers of this phenotype shift remains contentious [6, 7]. Evidence from human studies show that immobilization results in attenuated rates of MPS [5]. However, difficulty in direct assessment of MPB has precluded its routine quantification in humans, and thus our understanding of the MPB contributions to immobilization-induced muscle atrophy remains deficient. While stable isotope methodologies determine the net synthesis or degradation of protein content, it fails to characterise the synthesis of individual muscle proteins which precludes identification of their biological functions within the muscle cell.

Few studies have assessed changes in specific muscle protein content in response to periods of unilateral lower limb suspension [8] or bedrest in humans [9] and the available data are limited to identification of a small number of individual proteins. Contemporary proteomic methodologies that are quantitative in nature [10] can provide novel data on changes in a large number of specific muscle protein contents. Furthermore, the application of gene set enrichment analyses (GSEA) can enhance our understanding of the biological processes associated with immobilization-induced muscle atrophy. It also provides an extensive dataset for integration with other high throughput 'omics' technologies (i.e., RNA sequencing).

Factors such as age, sex, dietary intake and physical activity undoubtedly influence the physiological responses to limb immobilization. Therefore, a highly standardised pre-intervention period and dietary intake throughout the entire immobilization period is imperative to minimise variation due to lifestyle factors and provide a point of reference to which future therapeutic interventions can be compared. Furthermore, standardised conditions provide an optimal platform to interrogate the mechanisms underpinning muscle atrophy. Therefore, the aims of this study were to quantify: 1) changes in the human phenotype in response to 14 days of knee-brace immobilization in men with a standardised acute training history and dietary intake; and 2) changes in the skeletal muscle proteome after three and 14 days of knee-brace immobilization via quantitative untargeted Sequential Window Acquisition of All Theoretical Mass Spectra (SWATH) proteomics and subsequently detail the enriched gene sets according to Gene Ontology Biological Processes (GOBP) annotations.

## Materials and methods

### Participant characteristics

Based on previous work [4], a sample of 16 participants were required to determine changes in muscle cross sectional area (CSA) with 14 d of immobilization (alpha 0.05, power 0.95 G*Power 3.1.9.6). Eighteen healthy male participants were included in the present study; these participants are a sub-group from a larger study prospectively registered with the Australian

New Zealand Clinical Trials Registry (ACTRN12616001399482). Measured and reported participant characteristics at baseline can be found in Table 1. Participants were eligible for inclusion if they were between the ages of 20 and 40 years, male, and reported a consistent exercise history for the prior 6 m. Participants were excluded if they reported any recent (<6 months) injury requiring immobilization, or medical conditions that would place participants at increased risk during exercise. All participants were screened for risk factors/chronic disease status and were cleared of any medical factors or medications that might influence the outcomes of this study. Participants were recruited via electronically distributed and physical flyers between October 2016 and May 2017 and received an honorarium for participation. This study was approved by Bond University's Human Research Ethics Committee (0000015478) and all participants provided written informed consent prior to commencing the study.

## Study overview

The study was undertaken over an eight-week period (Fig 1). Participants initially completed a graded exercise test on a cycle ergometer and three repetition maximum (3RM) testing for prescription of the exercise to take place in the four-week exercise programme that preceded the immobilization protocol. Seven days prior to the first baseline muscle biopsy and start of immobilization, participants commenced a standardised, prescribed 21-d dietary intake that continued throughout the 14-d immobilization protocol. Physiological testing occurred over the three days before and after immobilization. During the immobilization period, participant's left knee joint was immobilized using a telescoping adjustable knee brace (X-Act ROM Knee Brace, DonJoy Orthopedics, TX, USA). The brace was individually fitted to fix the position of the knee joint at 60˚ flexion and secured with a single-use fastener. Participants were instructed not to weight bear on the immobilized limb and were provided forearm crutches for ambulation. Immediately prior to immobilization and at day three and 14 (end) of immobilization, muscle biopsies were obtained from the *vastus lateralis* as per prior proteome analyses [8, 9], under local anaesthetic (1% xylocaine) using a 5 mm Bergstrom needle with manual suction. Skeletal muscle was washed with ice-cold saline, separated from any visual connective tissue, and immediately snap-frozen in liquid nitrogen until further analysis.

## Dietary control

Participants were prescribed a diet to achieve energy balance based on the Schofield equation and a physical activity level (PAL) of 1.5 [12]. Diets were prescribed to provide 1.5 g/kg body

**Table 1. Participant characteristics at baseline (mean ± SD) prior to commencing the study.**

| | |
|---|---|
| Age (y) | 25.4 ± 5.5 |
| Height (cm) | 179.4 ± 5.2 |
| Body mass (kg) | 81.2 ± 11.6 |
| $VO_2$peak (mL·min$^{-1}$) | 3448.8 ± 684.4 |
| Peak aerobic power output (W) | 242.2 ± 49.8 |
| 3RM unilateral leg press (kg) | 118.1 ± 23.5 (CON) |
| | 116.4 ± 23.6 (IMM) |
| Aerobic training frequency (sessions·week$^{-1}$) | 1.6 ± 1.7 |
| Aerobic training volume (min·session$^{-1}$) | 46.0 ± 58.4 |
| Resistance training frequency (sessions·week$^{-1}$) | 2.4 ± 2.0 |
| Resistance training volume (min·session$^{-1}$) | 41.8 ± 34.5 |

CON = control limb; IMM = immobilized limb; $VO_2$peak = peak oxygen uptake; 3RM = three repetition maximum.

**Fig 1. Study overview.** Two primary testing phases were interspersed with a 14 d period of unilateral lower limb immobilization, and preceded by a 4 week exercise programme. Muscle biopsies were obtained from the *vastus lateralis* before, and at days 3 and 14 of immobilization. *3RM, 3 repetition maximum testing; Crutch icon, start of immobilization; GET, Graded exercise test [11]; DXA, Dual-energy X-ray absorptiometry; MRI, Magnetic resonance imaging; Muscle icon, muscle biopsy.*

mass of protein per day. All foods and drink were provided to participants in full and participants were not permitted to consume any additional food or calorie-containing fluid outside of their prescribed diet. Meals were provided by a commercial provider (24% protein; Lite n' Easy, Brisbane, Australia), with individualised supplementary food prescription overseen by an Accredited Practicing Dietitian to increase the protein and/or carbohydrate content on a daily basis. Participants were required to complete and return a daily checklist, to show all prescribed foods were consumed and no additional food was consumed. Habitual dietary intake of participants was not recorded in this study, given the comprehensive dietary provision prior to, and during, the immobilisation period.

## Unilateral 3RM leg press and aerobic power/VO$_2$peak

Three repetition maximum (3RM) testing was conducted on a plate-loaded unilateral leg press. Each repetition was completed to a hip and knee joint angle of 90° flexion. Warm up consisted of ten repetitions without external load, eight repetitions at an estimated 30%, and six repetitions at an estimated 60% 3RM, with 1 min rest between sets. After 3 min rest a 3RM was attempted. 3RM was achieved within three attempts with 3 min rest between attempts.

Participants completed a maximal graded exercise test to determine peak aerobic power output (PPO) and maximum oxygen uptake (VO$_2$ peak) on a cycle ergometer. Participants commenced cycling at a work-rate of 100 W for 150 s. Work rate increased by 50 W for the next 150 s, and 25 W every 150 s thereafter until volitional exhaustion [11]. Testing was conducted on an Excalibur Sport ergometer (Lode, Groningen, Netherlands) and expiratory gasses analyzed by metabolic cart calibrated to manufacturer's instructions (CosMed, Rome, Italy). All physical testing and training were facilitated by an Exercise Scientist or Accredited Exercise Physiologist at the Bond Institute of Health and Sport facilities.

## Muscle volume (MRI) and lean mass (DEXA)

Participants reported to the Bond Institute of Health and Sport for anthropometric procedures (0600–0800 h) after an overnight fast. A dual energy x-ray absorptiometry scan (DEXA; Lunar Prodigy, GE Healthcare, Madison, WI, USA) was administered by a licensed operator. Analysis of scans was subsequently completed with regions of interest adjusted as required (GE encore 2016 software, GE Healthcare, Madison, WI, USA).

Participants then immediately reported to Queensland Diagnostic Imaging for Magnetic resonance imaging (MRI; 3T Magnetom Skyra, Siemens Healthineers, Victoria, Australia) of the left thigh, with a fish oil capsule used as the marker at the mid-thigh landmark that was located pre-immobilization and marked with permanent marker to assist with replication in

post-immobilization testing. Five × 5 mm slices were imaged from proximal to distal with 2 mm gaps (field of view = 380 mm; resolution = 336 × 448). Imaging was performed with participants supine and heels fixed to standardise the distance of separation knees supported to control the joint/scan angle. During post-immobilization scanning the knee-brace was briefly removed but participants did not weight bear at any point moving to or from the scanning bed, and the knee-brace was promptly reapplied after scanning. Computation of muscle cross sectional area (CSA; cm$^2$) for the *vastus group*, *rectus femoris* and *quadriceps femoris* was performed on the middle slice by manual tracing using (OsiriX Lite 8.0, Pixmeo, Bernex, Switzerland) [13].

## Exercise programme

Participants completed a four-week exercise programme prior to knee-brace immobilization, consisting of four exercise sessions per week, alternating between resistance and cycling exercise. Resistance programs contained upper- and lower-body compound and isolation exercises, with exercise intensity ranging from 60–80% predicted 1RM with 2–3 sets of 8–12 repetitions. Cycle training consisted of one 30-min steady-state (60% PPO) bout, and one 30-min interval training session, containing three × 3-min intervals at 65–70% PPO. All participants completed the same relative training load for each exercise session.

## Proteome analysis

**Muscle preparation.**    All procedures for proteomics were undertaken at the Australian Proteome Analysis Facility, Macquarie University, Australia. The sample preparation of human skeletal muscle was performed using modified methods that are described by Mirzaei and colleagues [14]. Human skeletal muscle was homogenised with 8 M urea, 100 mM Tris-HCl (pH 8) using Precellys tissue homogeniser (Bertin Instruments). Samples were lysed using 3 × 20s cycles. The supernatant was collected and centrifuged to remove any debris. The protein concentration for each sample is determined by BCA assay using bovine serum albumin as a standard. Samples were diluted (1:10) in 50 mM Tris-HCl and then 35 μg of protein from each sample was taken for digestion, and the final volume adjusted to 50 μL using 50 mM Tris-HCl. Samples were reduced with dithiothreitol (10 mM), alkylated with iodoacetamide (25 mM) followed by digestion firstly with Lys-C (Wako, Japan) at a 1:100 enzyme-protein ratio for three hours at room temperature, and further digested with Trypsin (Promega, USA) at a 1:100 enzyme-protein ratio for 16 hours at 37˚C. Following digestion, pH was adjusted to 3 using a final concentration of 1% TFA, and each sample desalted using stage tips containing Styrene Divinyl Benzene (Empore SDB-RPS 47 mm extraction disk, Supelco). Briefly, stage tips were self-packed into pipette tips, peptides were bound to the stage-tip, washed with 0.2% TFA and eluted with 80% (v/v) acetonitrile, 5% (v/v) ammonium hydroxide. The cleaned peptides were dried using a vacuum centrifuge and reconstituted in 35 μL of loading buffer (2% (v/v) acetonitrile, 0.1% (v/v) formic acid, 97.9% (v/v) water). For SWATH-MS data, 4 μL of the digested sample was taken and diluted with the loading buffer to a final volume of 10 μL prior to injection. SWATH was acquired in random with a blank run in between each sample.

**High pH reverse phase-HPLC (HpH RP-HPLC).**    For ion library generation through high pH fractionation, a pool was prepared from each digested sample and fractionated by HpH RP-HPLC. The sample was resuspended in mobile phase buffer A containing 5 mM ammonium hydroxide solution (pH 10). The composition of buffer B was 5 mM ammonia solution with 90% Acetonitrile (pH 10). After sample loading and washing with 3% buffer B for 10 mins at a flow rate of 300 μL/min, the buffer B concentration was increased from 3% to 30% over 55 mins and then to 70% between 65 to 75 mins and to 90% between 75–80 mins.

The eluent was collected every 2 mins at the beginning of the gradient and at 1 min intervals for the rest of the gradient.

**2D-IDA.** Following HpH-RP-HPLC separation, 18 fractions were concatenated (0–82 min), dried and resuspended in 25 μL of loading buffer. 10 μL/fraction was taken for 2D-IDA analysis.

**Information dependent acquisition (IDA) and SWATH acquisition.** A 6600 TripleTOF mass spectrometer (Sciex, Framingham, MA) coupled to an Eksigent Ultra-nanoLC-1D system (Eksigent Technologies, Dublin, CA) was employed for both IDA and SWATH-MS analysis. Peptides were loaded onto a reverse phase peptide C18 self-trap (Halo-C18, 160 Å, 2.7 um, 200 μm × 10 mm) for pre-concentration and desalted for 3 min with the loading buffer at a flow rate of 5 μL/min. After desalting, the peptide trap was switched in-line with an analytical column (15 cm × 200 μm, nano cHiPLC column (ChromXP C18-CL 3 μm particles—120 Å pores)). Peptides were eluted and separated from the column using the buffer B (99.9% (v/v) acetonitrile, 0.1% (v/v) formic acid) gradient starting from 5% and increasing to 35% over 120 min at a flow rate of 600 nL/min. After peptide elution, the column was flushed with 95% buffer B for 6 min and re-equilibrated with 95% buffer A (2% (v/v) acetonitrile, 0.1% (v/v) formic acid, 97.9% (v/v) water) for 10 min before next sample injection. In IDA mode, a TOFMS survey scan was acquired at m/z 350–1500 with 0.25 sec accumulation time, with the twenty most intense precursor ions (2+–5+; counts > 200 counts/second) in the survey scan consecutively isolated for subsequent product ion scans. Dynamic exclusion was used with a window of 30 sec. Product ion spectra were accumulated for 100 milliseconds in the mass range m/z 100–1800 with rolling collision energy. For SWATH-MS, identical LC conditions were used as described above, with m/z window sizes determined based on precursor m/z frequencies in previous IDA data. SWATH variable window acquisition with a set of 100 overlapping windows (1 amu for the window overlap) was constructed covering the mass range of m/z 399.5–1249.5. In SWATH mode, first a TOFMS survey scan was acquired (m/z 350–1500, 0.05 s) then the 100 predefined m/z ranges were sequentially subjected to MS/MS analysis. Product ion spectra were accumulated for 30 milliseconds in the mass range m/z 350–1500.

**IDA and SWATH data analysis.** Protein identifications from 2D-IDA data were performed with ProteinPilot (v5.0, Sciex) using the Paragon algorithm in thorough mode. The search parameters were as follows: sample type: identification; cys alkylation: iodoacetamide; digestion: trypsin + Lys C; instrument: TripleTOF 6600; special factors: none; species: Homo sapiens; ID focus: biological modifications. The database used was obtained from SwissProt (20,386 entries, August 2018). A reversed-decoy database search strategy was used with ProteinPilot, with the calculated protein FDR equalling 1%. The ProteinPilot group file from the 2D-IDA search result was imported into PeakView (v2.2; Sciex) and used as a local peptide assay library. This library contained 3208 identified proteins (S1 Table in S1 File). SWATH peaks were then extracted using PeakView. Shared and modified peptides were excluded. Peak extraction parameters were set as the following: 100 peptides per protein, 6 transition ions per peptide, peptide confidence threshold 99%, FDR extraction threshold 1%, Extract Ion Chromatogram retention time window 5 min and mass tolerance 75 ppm. The extracted transition ion peak areas, peptide peak areas and protein peak areas were exported for further statistical analysis.

## Statistical and bioinformatics analysis

All phenotype data are presented as mean ± standard deviation, or box and whisker plot representing the median (and mean) and 25th to 75th percentiles, and ranges. Phenotype data were analysed by two-way or one-way repeated measures analysis of variance or non-parametric

equivalent, and Šídák's multiple comparisons tests were used to explain time and/or group effects with alpha adjusted for multiple comparison. Normality of phenotype data was assessed by Shapiro-Wilk test. A paired t-test (for quadriceps femoris and vastus group) and Wilcoxon matched pairs test (for rectus femoris due to non-normal distribution) was used for pre- to post-immobilization (MRI) comparisons, and alpha was set at $P < 0.05$ for all analyses (Graph-Pad Prism version 9.0.0 for MacOS, GraphPad Software, California USA).

Quantitative MS data was obtained from 54 samples across three time points. Extracted SWATH protein peak areas were analysed in R/Bioconductor using SwathXtend [15] to report on differentially expressed proteins. Differential expression analysis using linear modelling and empirical Bayes methods was carried out in limma v3.46 [16]. Paired sample comparisons were employed using participant ID as a blocking factor in the design matrix, to compare immobilized (3 d or 14 d) and baseline protein levels within participants. A false discovery rate (FDR) was applied to correct for multiple comparisons, with statistical significance accepted at an FDR <0.05. For gene set enrichment analysis (GSEA), UniProt IDs were first ranked using the signed moderated t-statistic from limma and interrogated against the Gene Ontology Biological Processes (GOBP) gene-sets using clusterProfiler v3.18.1 package (10,000 permutations; gene set size range 25–500) [17]. Gene set enrichments were visualized as networks in Cytoscape v3.8 using the EnrichmentMap v3.3.1 package under conservative thresholds ($P < 0.001$, FDR<0.05 and a combined similarity cut-off > 0.325) [18]. Groups of like terms were summarised (two or more gene-sets per cluster) using the AutoAnnotate v1.3.3 package, and the most statistically significant GOBP term in each cluster plotted together [19]. To investigate specific GOBP terms at the protein level, co-expression networks were visualized using the GeneMANIA package [20]. Hub proteins were defined as proteins with the highest 5% of connectivity in their respective co-expression network. The mass spectrometry proteomics data have been deposited to the ProteomeXchange Consortium via the PRIDE [21] partner repository with the dataset identifier PXD034908.

## Results

### Dietary intake

All participants adhered to dietary standardisation as assessed by completed diet diaries. Total energy intake was 146.6 ±10.4 kJ.kg$^{-1}$.day$^{-1}$, containing 1.47 ±0.05 g.kg$^{-1}$.day$^{-1}$ protein, 5.81 ±0.68 g.kg$^{-1}$.day$^{-1}$ carbohydrate and 0.74 ±0.06 g.kg$^{-1}$.day$^{-1}$ fat.

### Unilateral 3RM leg press and aerobic power/VO$_2$peak

There was a limb × time interaction for unilateral 3RM leg press strength (P <0.0001). There was a decrease in 3RM for the immobilized limb between pre- and post-immobilization (-20.28 ±11.85 kg, P <0.0001), with no change in the control limb (1.39 ±9.20 kg, P = 0.83; Fig 2A). The 3RM was lower for the immobilized compared to control limb at the post-immobilization timepoint (-22.08 ± 13.35 kg, P = 0.03). There was an effect of time for VO$_2$peak (mL.min$^{-1}$; P = 0.049) and PPO, with VO$_2$peak decreasing -2.88 ±5.19% (P = 0.044) and PPO -5.83 ±3.08% (P <0.0001) from pre- to post-immobilization.

### Lean mass (DEXA) and muscle CSA (MRI)

There was a main effect of time for leg lean mass (P = 0.0008), with a decrease in leg lean mass for the immobilized limb between pre- and post-immobilization (-0.33 ±0.40 kg, P = 0.0024) but no change in the control limb (-0.16 ±0.38 kg, P = 0.19; Fig 2B). Muscle CSA (cm$^2$) decreased from pre- and post-immobilization for the quadriceps femoris (-8.51 ±4.24%, P

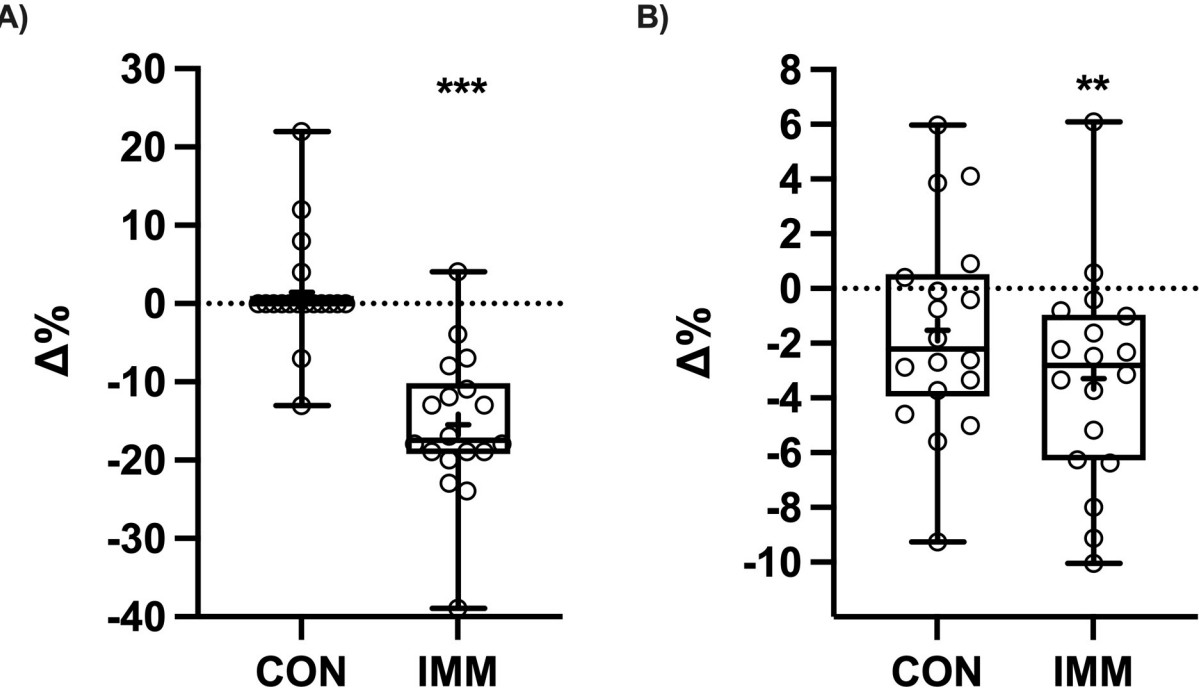

**Fig 2.** Phenotype change in control (CON) and immobilized (IMM) limb after 14 days of knee brace immobilization showing percent change in (A) three repetition maximum (3RM) unilateral leg press and (B) leg lean mass assessed by DEXA (n = 18). Data were analysed using two-way repeated measures ANOVA. **P <0.01, ***P <0.001 for Šídák's multiple comparison test (Pre- to Post-immobilization). +represents mean. Box contains the median (line) and shows 25th to 75th percentile, and whiskers represent minimum and maximum values.

<0.0001; Fig 3A) and vastus group (-9.56 ±4.60%, P <0.0001; Fig 3B). There was no change in muscle CSA for rectus femoris (0.24 ±7.07%, P = 0.90; Fig 3C). There was also a pre- and post-immobilization decrease in total muscle (-2.84 ±3.09%, P = 0.0009) and thigh circumference (-2.21 ±1.43%, P <0.0001).

## Proteome changes and gene set enrichment analysis (GSEA)

A total of 2281 quantifiable proteins were consistently identified by SWATH-MS. Differential expression analysis revealed changes in only two proteins following three days of limb immobilization (S2 Table in S1 File). However, 99 (76 up and 23 down) proteins were different to baseline at day 14 (FDR <0.05; Fig 4; S3 Table in S1 File). Table 2 outlines the 10 most reliably changed (lowest FDR) positively and negatively enriched proteins after 14 d of immobilization compared to baseline.

Following GSEA, no GOBPs were altered after three days of immobilization compared to baseline (S4 Table in S1 File). However, at day 14, 322 (263 positive and 59 negative) gene-sets were enriched (FDR <0.05, P <0.001; S5 Table in S1 File). To reduce redundancy in reporting, these GOBPs were grouped with like terms into 24 (20 positive and 4 negative) clusters (S6 Table in S1 File). Fig 5 characterises the grouped processes and identifies the number of GOBPs within each of the 24 clusters, ranked by the largest NES within that cluster.

Co-expression networks to understand the individual protein enrichment patterns among GOBPs of interest found to be significantly positively/negatively enriched after 14 d of immobilization (Fig 6). These networks were mapped to interrogate mechanisms of interest based on prior work showing mitochondrial dysfunction [2, 9], ROS production and antioxidant defences systems [8, 22] and proteolysis [22] all implicated in the muscle atrophy response.

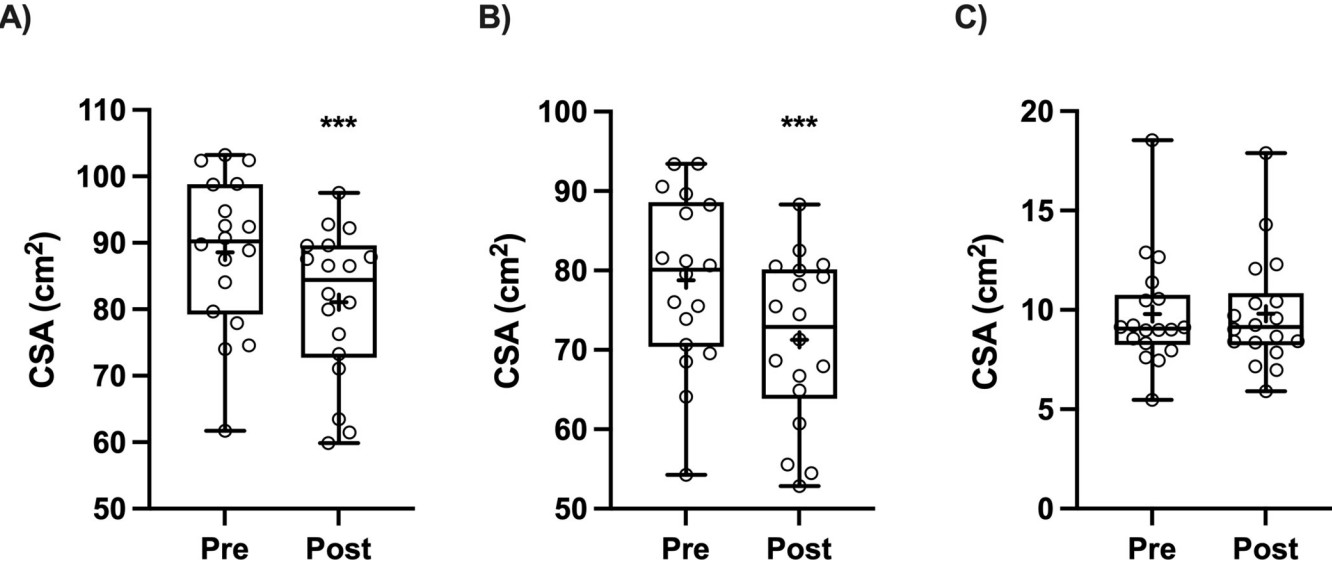

**Fig 3.** Pre-immobilization and post-immobilization (14 d) cross sectional area (cm$^2$) of immobilized limb (A) *quadriceps femoris*, (B) *vastus group*, and (C) *rectus femoris* assessed by MRI (n = 18). Data were analysed using paired t-tests and Wilcoxon matched-pairs test (rectus femoris). ***P <0.001 compared to Pre-immobilization. +represents mean. Box contains the median (line) and shows the 25th to 75th percentile, and whiskers represent minimum and maximum values.

## Discussion

This study characterises the skeletal muscle loss and associated proteome response to 14 days of unilateral knee-joint immobilization in young men. The main findings were that: 1) minimal changes (2 proteins) in protein content were observed after 3 d, but a range of proteins were differentially expressed (99 proteins) compared to baseline protein content after 14 d of immobilization; 2) 322 GOBPs were differentially enriched after immobilization including mitochondrial organisation, cellular stress and proteasome proteolysis; and 3) the decrease in muscle cross-sectional area, limb lean mass and peak oxygen uptake were similar to those

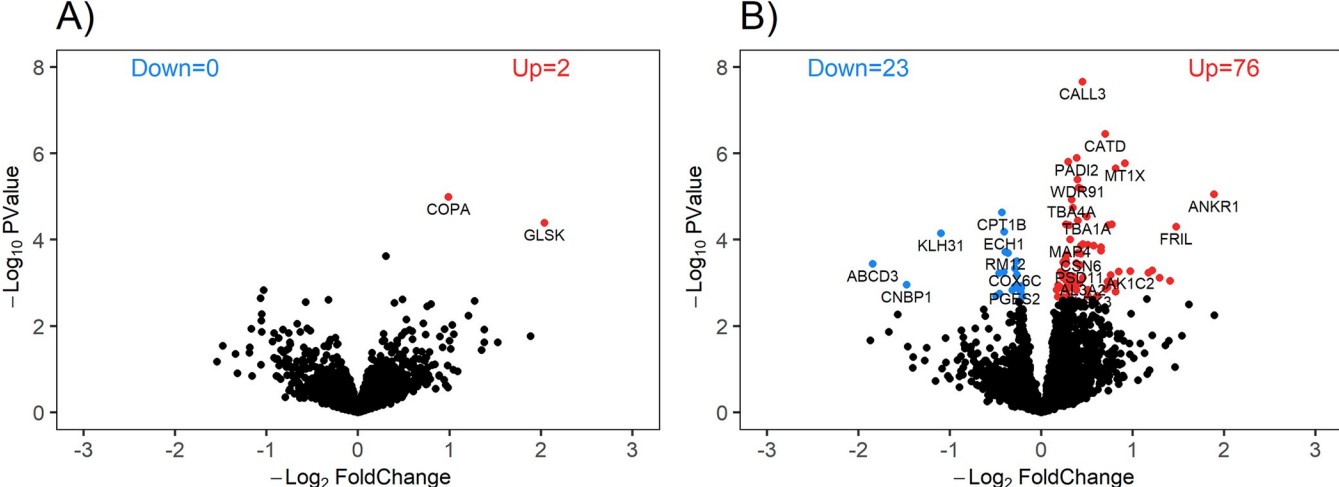

**Fig 4.** Volcano plot at (A) day 3 and (B) day 14, compared to baseline. Coloured datapoints represent proteins positively (red) and negatively (blue) enriched proteins meeting a threshold FDR<0.05.

**Table 2. Ten most reliably changed (lowest FDR) positively and negatively enriched proteins after 14 d of immobilization compared to baseline.**

| Uniprot ID | Symbol/Gene | Protein name | Fold change | FDR |
|---|---|---|---|---|
| | | **Positively enriched proteins** | | |
| P0DP25 | CALM3 | Calmodulin 3 | 1.39 | 1.28E-05 |
| P27482 | CALL3 | Calmodulin-like protein 3 | 1.37 | 2.52E-05 |
| P07339 | CATD | Cathepsin D | 1.63 | 2.71E-04 |
| Q9Y2J8 | PADI2 | Protein-arginine deiminase type-2 | 1.31 | 6.47E-04 |
| P55072 | TERA | Transitional endoplasmic reticulum ATPase | 1.23 | 6.47E-04 |
| P80297 | MT1X | Metallothionein-1X | 1.88 | 6.47E-04 |
| Q9Y394 | DHRS7 | Dehydrogenase/reductase SDR family member 7 | 1.76 | 7.34E-04 |
| A4D1P6 | WDR91 | WD repeat-containing protein 91 | 1.31 | 1.15E-03 |
| P35080 | PROF2 | Profilin-2 | 1.33 | 1.50E-03 |
| P28289 | TMOD1 | Tropomodulin-1 | 1.35 | 1.50E-03 |
| | | **Negatively enriched proteins** | | |
| Q92523 | CPT1B | Carnitine O-palmitoyltransferase 1 | -1.35 | 3.80E-03 |
| Q13011 | ECH1 | Delta(3,5)-Delta(2,4)-dienoyl-CoA isomerase, mitochondrial | -1.32 | 6.87E-03 |
| Q9H511 | KLH31 | Kelch-like protein 31 | -2.14 | 7.09E-03 |
| P52815 | RM12 | 39S ribosomal protein L12, mitochondrial | -1.31 | 1.41E-02 |
| P35613 | BASI | Basigin | -1.29 | 1.42E-02 |
| P45880 | VDAC2 | Voltage-dependent anion-selective channel protein 2 | -1.20 | 1.88E-02 |
| P28288 | ABCD3 | ATP-binding cassette sub-family D member 3 | -3.59 | 1.99E-02 |
| P09669 | COX6C | Cytochrome c oxidase subunit 6C | -1.22 | 2.34E-02 |
| Q9Y277 | VDAC3 | Voltage-dependent anion-selective channel protein 3 | -1.33 | 2.57E-02 |
| Q86SX6 | GLRX5 | Glutaredoxin-related protein 5, mitochondrial | -1.37 | 2.61E-02 |

reported in previous studies, despite prolonged standardisation of exercise training and dietary intake prior to immobilization.

This is the first study to use SWATH-MS quantitative proteomics [10] to assess the skeletal muscle proteome response to unilateral knee-brace immobilization. After 3 d of immobilization, we found only two proteins that were differentially expressed compared to baseline. Our data show the Coatomer subunit alpha (COPA) protein implicated in intracellular protein transport, and the enzyme Glutaminase (GLS) purported to regulate metabolism of glutamine to glutamate [23], were upregulated after three days of muscle unloading compared to baseline (FC: 1.99 and 4.11, respectively). Given skeletal muscle is purported to contain up to 80% of body glutamine, is the key donor of glutamine to plasma [24] and integral to immune cell function during stress responses [23], the upregulation of the enzyme responsible for glutamine hydrolysis is of interest. Potentially, an increase in cellular glutamine content as a result of early proteolysis [25] may have driven an increase in observed GLS content in muscle. Whether or not the upregulation in GLS content was also required to facilitate intermediates to the tricarboxylic acid cycle in response to cessation of contractile activity and atrophy related energy-stress is speculative. Nonetheless, our findings show minimal protein permutation in the early phase of limb immobilization. Indeed, the early cellular response to immobilization is more likely to up/down regulate transcription given the time-course for changes in specific functional proteins [26, 27], and prior work has shown little change in mitochondrial enzymes after 5 d of immobilization [28]. Consequently, given the limited changes in the muscle proteome after only three days immobilization we primarily focused on differences in protein content at day 14.

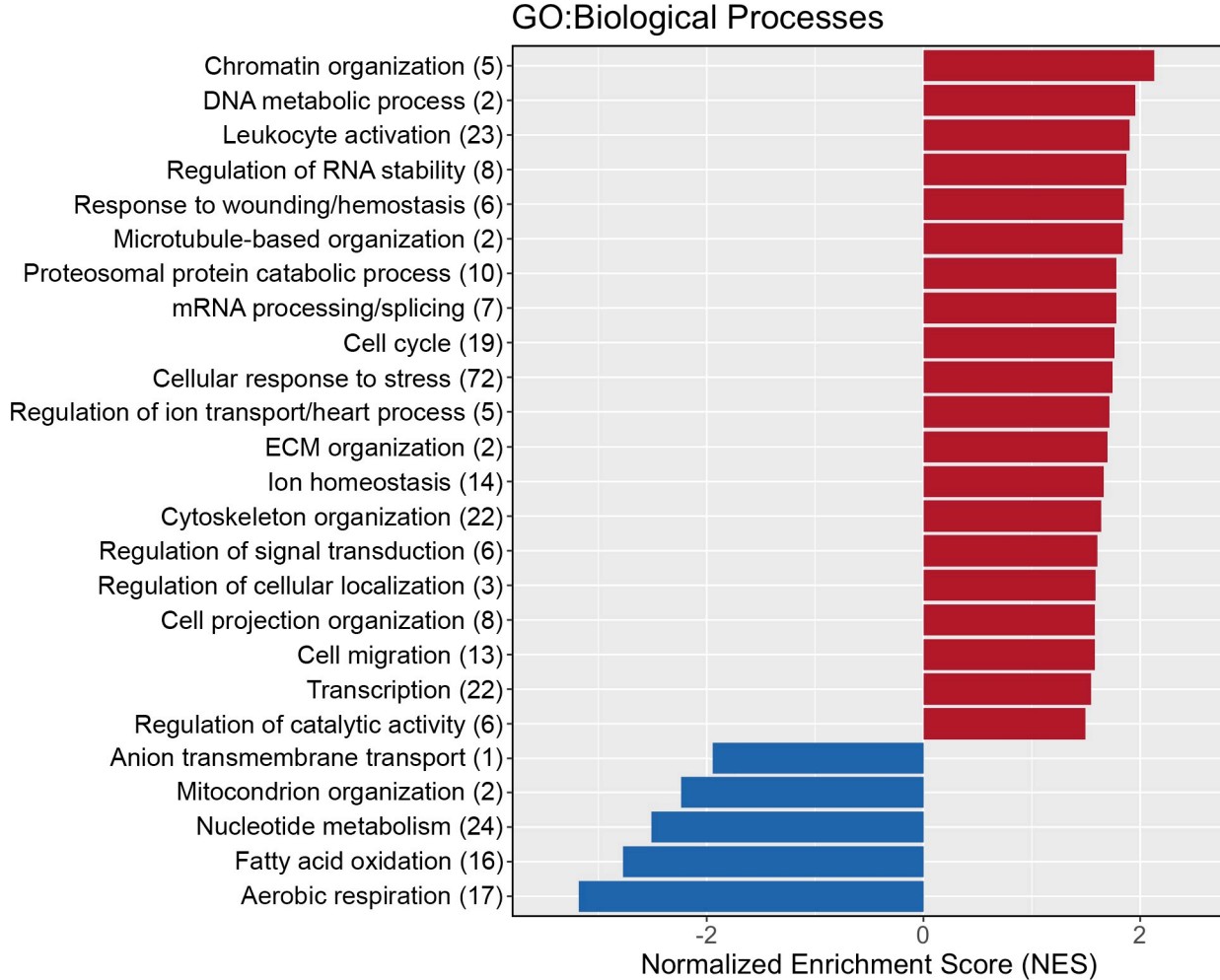

**Fig 5. Characterisation of the 24 clusters of Gene Ontology Biological Processes (GOBP) formed.** Title of bars are representative terms for each cluster, and the number of GOBPs within each cluster are represented in parenthesis. Colour represents direction of change in expression of the majority of GOBPs within that cluster compared to baseline (red = positive enrichment, blue = negative enrichment). The magnitude of bars represents the largest normalised enrichment score of a GOBP within that cluster.

After 14 d of immobilization, we identified 99 proteins that were differentially expressed relative to baseline levels at a false discovery rate of < 5%. These data complement and extend on previous studies employing analysis of individual proteins. For example, Abadi and colleagues [2] report a decreased (~20%) protein content of COX2, and reduced activity of COX and CS in young men (n = 12, 21 ±2 y) and women (n = 12, 21 ±3 y) after 14 d of immobilization. Microarray analysis also showed a decrease in transcriptional activity of intermediaries for carbohydrate metabolism. Our data show a reduction (FDR <0.05) in CS protein content and a decrease in other novel enzymes and proteins within the tricarboxylic acid cycle (MDH2, IDH3B), electron transport chain (NDUFV1, NDUFV2, NDUFA5, COX6C) and beta oxidation process (ECH1, CPT1B), as well as the mitochondrial protein PERM1. Accordingly, our data agree with previous studies and provides new information in support of lowered muscle protein content within multiple energy deriving pathways, and the down regulation in mediators of mitochondrial biogenesis [29] in response to immobilization.

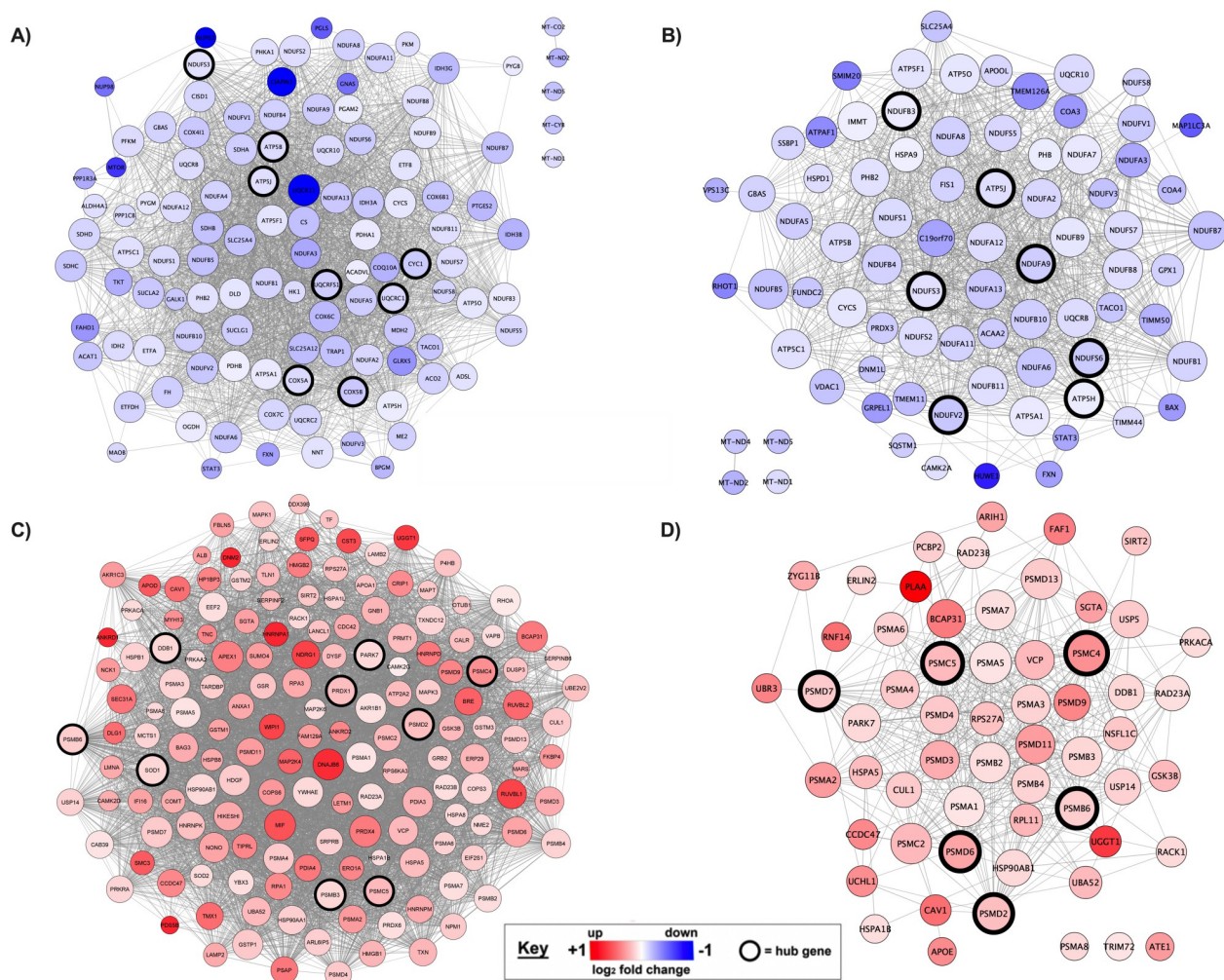

**Fig 6.** Co-expression networks for gene sets A) *Generation of precursor metabolites and energy* (GO:0006091), B) *Mitochondrion organization* (GO:0007005), C) *Cellular response to stress* (GO:0033554), and D) *Proteasomal protein catabolic process* (GO:0010498). Nodes correspond to individual proteins enriched at day 14 compared to baseline and edge lines between two proteins represent a co-expression relationship. Colour represents direction of change in expression compared to baseline (red = positive enrichment, blue = negative enrichment) and intensity of colour represents the magnitude of change (darker = higher fold change). Black borders surrounding nodes represent 'hub' proteins characterized by the highest 5% of connectivity within the respective gene set.

Brocca and colleagues [8] have previously examined the skeletal muscle proteome response to three weeks of limb immobilization, using two-dimensional electrophoresis for protein separation and MALDI-ToF-MS for protein identification [8]. This study [8] reported 43 differentially expressed protein spots, resulting in the identification of 25 unique and differentially expressed proteins after three weeks of unilateral lower limb suspension in a cohort of eight young men. A limitation associated with these [8] methods are the lower number of identified proteins and limited quantitative capacity compared to contemporary proteomics assessments; as a result, comparisons with the present study are limited to comparisons of individual protein contents and overarching themes. Brocca and colleagues [8] categorised differentially expressed proteins in their study as those relating to energy metabolism (downregulated) and antioxidant defence systems (upregulated). We identified all proteins identified by Brocca and colleagues, indicating robust replication of the physiological response to immobilization and validation of our SWATH methodology. The SWATH methodology unique to our study

permitted not only a greater yield of differentially expressed proteins, but use of GSEA, and this is the first study to provide a more comprehensive description of the biological processes impacted by limb immobilization [30].

Our GSEA identified 59 biological processes with a reduced protein content compared to baseline after 14 d of immobilization, and these terms were grouped into four unique clusters. As such, the atrophy response in our study was characterized by reduced mitochondrial organization and energy metabolic processes (Fig 5). Collectively, it appears energy-deriving biological processes are attenuated in muscle with an extended period of immobilization. Individual biological processes and proteins may reveal key regulators of energy metabolism dysfunction. For example, we identified hub proteins with potential for being central regulators for the *generation of precursor metabolites and energy (GO:0006091)* and *mitochondrion organization (GO:0007005*; Fig 6) due to their high connectivity within these networks, despite many of these proteins having higher false discovery rates (FDR >0.05) than typically acceptable when considering individual pairwise comparisons.

Hub proteins for the *generation of precursor metabolites and energy* included two mitochondrial membrane ATP synthases and multiple components of respiratory chain complex I, III and IV. There was some overlap in the hub proteins identified for *mitochondrion organization* (NDUFS3, ATP5J/ATP5PF), but briefly we identified five core/accessory subunit of respiratory chain complex I and two ATP synthases as proteins with the highest connectivity within this network. Of note, the decrease in protein content within these metabolic and energetic biological processes was associated with a reduced aerobic capacity. Whether or not the content of these proteins are negatively enriched due to a reduction in demand, or whether they are targeted for degradation to prevent mitochondrial dysfunction is unclear [31]. Regardless, reductions in the capacity for energy production will have negative implications on other energy-dependant biological processes. It should be noted however, that our data represent protein changes within the muscle cell, but the organ systems or whole-body metabolic consequences are yet to be fully determined [32]. Although recent work has examined the plasma metabolome of older men following 7 days of post-surgery bedrest [33], the vastly different population groups prevent meaningful comparison to our work. To date, no study has examined the metabolome in response to unilateral lower limb immobilization, and future work should employ untargeted metabolomics to gain further insight to the wider metabolic impact of these cellular responses. Nonetheless, we have identified key mitochondrial proteins that are negatively enriched in skeletal muscle after 14 d of unilateral lower limb immobilization, that likely reflect the reduced requirement for energy metabolism and a catabolic cellular environment.

Previous work has identified changes in mitochondrial morphology and function during periods of muscle disuse and ensuing ROS production [31, 34, 35]. We found a negative enrichment in *inner mitochondrial membrane organization (GO:0007007)* at the pathway level, which is not directly representative of changes in morphology *per se*, but may be indicative of disruption to mitochondrial protein trans-membrane transportation. The data also show that the mitochondrial protein VDAC2, an outer mitochondrial membrane protein for the diffusion of small molecules, was one of the most reliably downregulated proteins in our sample (Table 2). Changes in mitochondrion organization in the present study coincided with a downregulation in *oxidation-reduction process (GO:0055114; S6 Table in* S1 File, *cluster 7)* and upregulation in *cellular response to stress (GO:0033554; S6 Table in* S1 File, *cluster 1)*, suggesting a physiological response to attenuate oxidative stress-induced maladaptation with muscle disuse. Indeed, dysregulated oxidative stress responses have been reported to drive proteolytic responses [22] and our GSEA provides support for such a contention. However, mapping and subsequent interrogation of proteins within these networks and identification of common

proteins between networks is needed to better understand any potential interactions between mitochondrial dysfunction, the oxidative stress response, and upregulation of proteolytic systems. Nonetheless, immobilization generates substantial changes in mitochondrial and associated metabolic proteins in skeletal muscle, and induces a significant cellular stress response.

Our findings show that 263 biological processes/pathways were positively enriched (P <0.001 and FDR <0.05) at the protein level following limb immobilization (S5 Table in S1 File), and these were subsequently grouped into 20 clusters of like terms (Fig 5, S6 Table in S1 File). Cluster 12, containing 10 terms, was clearly characterized by proteolysis and included GOBPs such as *proteasomal protein catabolic process (GO:0010498)*. There is significant debate about the contribution of muscle protein breakdown to the prolonged decrease in net protein balance that generates muscle atrophy [6, 22, 26]. Recently, Willis and colleagues [26] conducted a short-term (4-d) immobilization protocol in a group of eight young men to assess the transcriptome derived molecular networks associated with reductions in deuterium-derived rates of muscle protein synthesis. They also report upregulation of ubiquitin dependant proteolytic processes at the *transcriptome* level, but suggest a targeted protein degradation of components within the protein-synthetic machinery, indirectly attenuates muscle protein synthesis rather than these mechanisms contributing to global proteolysis. Our data concur, in part, that degradation of the protein synthetic machinery may still be evident after 14 d given the significant downregulation of muscle 39S ribosomal protein content (Table 2). However, we also show upregulation of other proteolytic systems, with the protease Cathepsin D identified as the third most reliably changed protein in our samples of 2281 proteins at day 14 of immobilization. Of note, the decrease in quadriceps CSA during the present study exceeds muscle loss reported to be solely attributed to reductions in MPS [36]. Accordingly, it seems intuitive to suggest the magnitude of muscle loss induced by 14 d immobilization dictates at least some increases in proteolytic activity contributes to muscle degradation, and our proteome data support this reasoning. Indeed, the magnitude of proteolytic contribution to muscle atrophy is likely individual due to hereditary factors, and potentially training history. Our prior work in rodents selectively bred to be high or low responders to endurance training [37], shows that superior endurance training adaptation attenuated upregulation in ubiquitination biological processes and atrophy in plantaris muscle, compared to rodents who responded poorly to endurance training. In contrast to plantaris muscle, we found no increased ubiquitination biological processes in the predominantly type I fibre soleus muscle, indicating fibre type and exercise-induced muscle phenotype may affect the cellular response to immobilization.

The GSEA in our study also identified changes in a number of biological processes not previously characterized at the muscle protein level. For example, the biological processes of *protein-DNA complex assembly (GO:0065004)*, *chromosome (GO:0051276)* and *chromatin organization (GO:0006325)* were positively enriched after immobilization. It appears that 14 days of muscle disuse significantly modifies DNA-protein interactions, with subsequent implications for DNA binding and transcriptional activity [38]. Interestingly, structural remodelling of chromatin is an energy consuming process [39], and would likely induce additional cellular energy demands that occur concomitant with downregulation of proteins implicated in energy metabolism and mitochondrial organisation/function. Nonetheless, our data shows upregulation of proteins involved in chromatin and nucleosome assembly, indicating either increased protein abundance or concentration due to cellular remodelling. However, given the evolving knowledge of epigenetic control of cell processes, further work is needed to understand the implications of these protein-DNA interactions. Moreover, extensive discussion of each positively enriched biological process within the immobilization-induced proteome response is

beyond the scope of the present study but provides new information for future studies determining the molecular regulation of muscle atrophy.

Despite our approach to understand the biological processes contributing to immobilization induced atrophy in the present study, the study is not without limitations. First, similar to other studies of human skeletal muscle, our methodology provides cross-sectional "snapshots" of the molecular profile during the atrophy process [40]. The mechanisms contributing to the skeletal muscle atrophy response during prolonged immobilization may be at least biphasic [41], and consideration of this complexity and time-course of changes during maladaptation is important context for interpretation of our data obtained after 14 d immobilization. As a result, the precise time-course of protein changes and the molecular events contributing to "early" versus "late" atrophy are yet to be fully determined; understanding this time-course will require more frequent muscle sampling, potentially in combination with dynamic measurements of protein synthesis/ breakdown to fully contextualise the observed changes in protein content. In this regard, and to reduce the number of samples obtained from participants, our study did not employ a within-person control (i.e., proteomic comparison to non-immobilized leg). Therefore, our data cannot compare differences between immobilized and non-immobilized limbs at specific time points. Second, although this study identified significantly more muscle proteins than any other research that has characterized the atrophy response to unilateral lower limb immobilization, we acknowledge that *only* 2281 proteins were consistently identified, and this equates to only a fraction of the entire muscle proteome that governs cellular activity. Of interest, our SWATH methodology did not consistently identify well-characterized ubiquitin ligases TRIM63 (MuRF1) and FBXO32 (MAFbx), despite TRIM63 being identified in our local library created from 2D-IDA. Collectively, this suggests that some of these ubiquitin ligases were not present in all samples, or were lowly expressed, and did not reach the detection threshold for our SWATH analyses.

In conclusion, this is the first study to assess the skeletal muscle proteome by SWATH proteomics methodology in response to 14 days of knee-brace immobilization. We interrogated 2281 quantifiable proteins and mapped protein permutations against GOBPs to show 322 biological processes are altered within the muscle cell during muscle unloading. Prominent changes in biological processes included a negative enrichment in mitochondrial organisation, and positive enrichment in cellular responses to stress and protein catabolic processes, observed concurrently with a decrease in muscle mass. These data provide a platform for future studies, examining inflammatory/catabolic disease, or cellular ageing, to compare proteomic signatures to elucidate the cellular adaptive responses to these conditions. Moreover, concurrently employing multiple untargeted "omics" techniques across periods of unilateral knee-brace immobilization will improve our understanding of the time course and interplay between the muscle transcriptome, proteome, and metabolome. Regardless, our findings characterise co-expression network hub proteins that provide new information of potential protein regulators of "simple muscle atrophy", which will be important for future research to develop countermeasures to the debilitating effects of muscle loss with prolonged immobilization and bedrest.

## Supporting information

**S1 File.**
(XLSX)

## Acknowledgments

We would like to thank our participants for volunteering for this prolonged study, Dr. Robert Fassett (M.D.) for undertaking the muscle biopsy procedures, Dr. Damon Arezzolo and Katsu

Shike who assisted with data collection, and Queensland Diagnostic Imaging for undertaking the MRI procedures. Aspects of this research have also been facilitated by access to the Australian Proteome Analysis Facility supported under the Australian Government's National Collaborative Research Infrastructure Strategy (NCRIS).

## Author Contributions

**Conceptualization:** Kristen L. MacKenzie-Shalders, Kevin J. Ashton, Vernon G. Coffey.

**Data curation:** Thiri Zaw, Kevin J. Ashton, Vernon G. Coffey.

**Formal analysis:** Thomas M. Doering, Jamie-Lee M. Thompson, Thiri Zaw, Kevin J. Ashton.

**Funding acquisition:** Kevin J. Ashton, Vernon G. Coffey.

**Investigation:** Thomas M. Doering, Jamie-Lee M. Thompson, Boris P. Budiono, Kristen L. MacKenzie-Shalders, Vernon G. Coffey.

**Methodology:** Kristen L. MacKenzie-Shalders, Kevin J. Ashton, Vernon G. Coffey.

**Project administration:** Thomas M. Doering, Jamie-Lee M. Thompson, Boris P. Budiono, Vernon G. Coffey.

**Visualization:** Thomas M. Doering, Kevin J. Ashton.

**Writing – original draft:** Thomas M. Doering.

**Writing – review & editing:** Thomas M. Doering, Jamie-Lee M. Thompson, Boris P. Budiono, Kristen L. MacKenzie-Shalders, Thiri Zaw, Kevin J. Ashton, Vernon G. Coffey.

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
