## [Decision Letter · Decision Letter 0]

2 May 2022

PONE-D-21-39372The muscle proteome reflects changes in mitochondrial function, cellular stress and proteolysis after 14 days of unilateral lower limb immobilization in active young menPLOS ONE

Dear Dr. Doering,

Thank you for submitting your manuscript to PLOS ONE. After careful consideration, we feel that it has merit but does not fully meet PLOS ONE’s publication criteria as it currently stands. Therefore, we invite you to submit a revised version of the manuscript that addresses the points raised during the review process.

We look forward to receiving your revised manuscript.

Kind regards,

Suman S. Thakur, Ph.D

Academic Editor

PLOS ONE

Journal Requirements:

This study was funded by the Collaborative Research Network for Advancing Exercise

and Sport Science (CRN-AESS - 201202) scheme awarded by the Department of

Education and Training, Australia. 

This study was funded by the Collaborative Research Network for Advancing Exercise

and Sport Science (CRN-AESS - 201202) scheme awarded to VGC and KJA by the Department of Education and Training Australia. The funding body played no role in the study design, data collection and analysis, decision to publish, or preparation of the manuscript.

Reviewers' comments:

Reviewer's Responses to Questions

**Comments to the Author**

1. Is the manuscript technically sound, and do the data support the conclusions?

Reviewer #1: Yes

Reviewer #2: Partly

2. Has the statistical analysis been performed appropriately and rigorously? 

Reviewer #1: Yes

Reviewer #2: Yes

3. Have the authors made all data underlying the findings in their manuscript fully available?

Reviewer #1: Yes

Reviewer #2: No

4. Is the manuscript presented in an intelligible fashion and written in standard English?

Reviewer #1: Yes

Reviewer #2: Yes

5. Review Comments to the Author

Reviewer #1: Thank you for the invitation to review this manuscript.

The authors present the results of detailed muscle proteomic analysis of 14-day, unilateral, lower limb immobilisation, in a small cohort of healthy, younger, adult human men. Their experimental design standardised common sources of confounding variables. Their results add significant additional detail to previous studies.

General comments

1. The authors have defined the goals of their study; designed a thorough standardisation process to minimise confounders; performed standard anatomical and physiological measures; conducted detailed proteomic studies; presented their data in a clear and detailed fashion.

2. Without wishing to ask the authors to speculate or extrapolate their results, what, if anything, can this study add to our understanding of age-related atrophy (sarcopaenia) and / or muscle loss in cachexia?

3. Similarly, can the authors expand further in their discussion, how their results tie in with the currently level of understanding in muscle metabolomics - see for example https://www.ncbi.nlm.nih.gov/pmc/articles/PMC8706620/

4. Given the complexity of this topic may I encourage the authors to include a brief section on next steps in advancing our understanding?

Reviewer #2: In this manuscript, Doering et. al describe the changes in the tissue proteome in joint-immobilized lower limb muscles. They characterize gross changes in lower limb mass and strength (using 3-repetition leg press test) in a prospectively cohort of 18 healthy male participants. They then use a novel mass-spectrometry based method in muscle biopsies to quantify changes in protein levels after 14 days of unilateral limb immobilization. They highlight significantly enriched and depleted proteins and assign biological significance to their findings by running Gene Set Enrichment Analyses (GSEA).

This manuscript is exceptionally well-written, and is clear and concise. The materials and methods section is very well detailed and allows for replication of the study and data analysis. While I do think that the work is of high quality, there are some concerns that I believe should preclude publishing the article in its current form.

Major concerns:

- The authors provide ample data on the results of their differential (limma) analysis for protein abundance but do not supply the readers/reviewers with the raw data on which they ran the analysis (patient-level protein abundance). I believe that this could be a very valuable resource and should be provided as a supplementary data sheet.

- The authors claim that all of the participants were subject to similar dietary and training schedules in the pre-immobilization period. However, they do not mention any baseline characteristics for the participants. I think such information is also valuable to include and could give the readers a sense of the homogeneity among the cohort.

- There are major limitations to the study design that are understandable but not well-acknowledged in the limitations section of the discussion. 1) The recruitment "via electronically distributed and physical flyers between October 2016 and May 2017" could introduce a bias when it comes to the physical fitness and baseline characteristics of the participants. 2) A sensible control experiment would have been to run a similar analysis on the non-immobilized leg to compare and contrast. This should be at least stated in the discussion.

- The authors have published on a similar (but separate) topic in the following work "Effect of short-term hindlimb immobilization on skeletal muscle atrophy and the transcriptome in a low compared with high responder to endurance training model". Since transcriptomic data is available on hindlimb immobilized muscles, could the authors compare the findings on the protein level to those on the RNA level? No formal analysis is necessary, but the findings should be discussed.

Minor concerns/comments:

- Why was the vastus lateralis muscle used for muscle biopsy? Is there a specific rationale or is it just for ease of biopsy?

- Line 91: replace "preludes" with "precludes"

- Line 303: The authors mention that "A paired t-test or Wilcoxon matched pairs test" were used. Could the authors be a little more specific? If both were done, that it should be stated as such. If one was picked over the other according to the analysis, this should also be clarified.

6. PLOS authors have the option to publish the peer review history of their article (what does this mean?). If published, this will include your full peer review and any attached files.

Reviewer #1: **Yes: **Jonathan Ball

Reviewer #2: No

---

## [Author Response · Author response to Decision Letter 0]

3 Jul 2022

Editor comments 

Author response: 

We have been sure to abide PLOS ONE's style requirements and file naming conventions throughout, including the naming conventions of these revision files. The only minor amendment to be made was changing “Figure” to “Fig” throughout. 

“Figure” has been CHANGED to “Fig” throughout this manuscript. 

2. Thank you for stating the following in the Acknowledgments Section of your manuscript: This study was funded by the Collaborative Research Network for Advancing Exercise and Sport Science (CRN-AESS - 201202) scheme awarded by the Department of

Education and Training, Australia. 

Please note that funding information should not appear in the Acknowledgments section or other areas of your manuscript. We will only publish funding information present in the Funding Statement section of the online submission form. Please remove any funding-related text from the manuscript and let us know how you would like to update your Funding Statement. Currently, your Funding Statement reads as follows:

This study was funded by the Collaborative Research Network for Advancing Exercise

and Sport Science (CRN-AESS - 201202) scheme awarded to VGC and KJA by the Department of Education and Training Australia. The funding body played no role in the study design, data collection and analysis, decision to publish, or preparation of the manuscript.

Author response: 

Thank you for bringing this to our attention. We have removed the funding information from the Acknowledgments section of our manuscript. We do not wish to make any changes to the funding statement provided. 

The following text has been DELETED from line 627-629 of the track changed manuscript:

“This study was funded by the Collaborative Research Network for Advancing Exercise

and Sport Science (CRN-AESS - 201202) scheme awarded by the Department of

Education and Training, Australia.”

Author response: 

We are committed to providing open access to our raw data files, and do not wish to make any changes to the Data Availability statement provided. We have uploaded all raw data to the PRoteomics IDEntification (PRIDE) database. The uploaded data includes all:

• IDA files for library generation (*.wiff and *.wiff.scan)

• library files from IDA searches (*.txt)

• SWATH data files (*.wiff and *.wiff.scan)

 Data is available under Project accession: PXD034908 (this will be available/released upon publication according to PRIDE).

The following text has been ADDED to line 337-339 of the track changed manuscript: 

“The mass spectrometry proteomics data have been deposited to the ProteomeXchange Consortium via the PRIDE (Perez-Riverol et al., 2022) partner repository with the dataset identifier PXD034908.”

Reviewer 1

The authors present the results of detailed muscle proteomic analysis of 14-day, unilateral, lower limb immobilisation, in a small cohort of healthy, younger, adult human men. Their experimental design standardised common sources of confounding variables. Their results add significant additional detail to previous studies.

General comments

1. The authors have defined the goals of their study; designed a thorough standardisation process to minimise confounders; performed standard anatomical and physiological measures; conducted detailed proteomic studies; presented their data in a clear and detailed fashion.

Author response: 

We appreciate your concise and favourable assessment of our work. 

No changes have been made to the manuscript. 

2. Without wishing to ask the authors to speculate or extrapolate their results, what, if anything, can this study add to our understanding of age-related atrophy (sarcopaenia) and / or muscle loss in cachexia?

Author response: 

As the first study to utilise SWATH proteomic analyses with GSEA to characterise the muscle proteome in response to muscle unloading in otherwise healthy individuals, we have been careful not to overstate our findings. Our work provides unique information about changes in the muscle cell at the protein level, in response to “simple muscle atrophy”, without confounding localised or systemic inflammatory or catabolic disease, or cellular ageing. Indeed, these additional confounding factors associated with disease and ageing will likely alter the proteomic signature associated with the muscle atrophy observed. As a result, not only does our data provide new information on the atrophic proteome response, it is also a platform for future works examining age-related atrophy (sarcopaenia) and / or muscle loss in cachexia, to compare proteomic signatures; differences between these proteomic signatures may provide further insight into the cellular adaptive responses to these conditions. We have provided a brief commentary to this affect within our discussion.

The following text has been ADDED to line 615-617 of the track changed manuscript: 

“These data provide a platform for future studies, examining inflammatory/catabolic disease, or cellular ageing, to compare proteomic signatures to elucidate the cellular adaptive responses to these conditions.”

3. Similarly, can the authors expand further in their discussion, how their results tie in with the currently level of understanding in muscle metabolomics - see for example https://www.ncbi.nlm.nih.gov/pmc/articles/PMC8706620/

Author response: 

Thank you for this suggestion. To date, no study has employed untargeted metabolomics to evaluate the cellular metabolic consequences of unilateral lower limb immobilization. In line with your comment, we have provided a brief commentary around the potential benefit of concurrent measurements of the metabolome, into our discussion. 

The following text has been ADDED to line 508-515 of the track changed manuscript: 

“It should be noted however, that our data represent protein changes within the muscle cell, but the organ systems or whole-body metabolic consequences are yet to be fully determined (Alldritt et al., 2021). Although recent work has examined the plasma metabolome of older men following 7 days of post-surgery bedrest (Kemp et al., 2020), the vastly different population groups prevent meaningful comparison to our work. To date, no study has examined the metabolome in response to unilateral lower limb immobilization, and future work should employ untargeted metabolomics to gain further insight to the wider metabolic impact of these cellular responses.”

4. Given the complexity of this topic may I encourage the authors to include a brief section on next steps in advancing our understanding?

Author response: 

Thank you for your recommendation. We have provided a brief comment on how we believe future work should progressed to best advance our understanding of simple muscle atrophy. 

The following text has been ADDED to line 618-620 of the track changed manuscript: 

“Moreover, concurrently employing multiple untargeted “omics” techniques across periods of unilateral knee-brace immobilization will improve our understanding of the time course and interplay between the muscle transcriptome, proteome, and metabolome.”

Reviewer 2 

In this manuscript, Doering et. al describe the changes in the tissue proteome in joint-immobilized lower limb muscles. They characterize gross changes in lower limb mass and strength (using 3-repetition leg press test) in a prospectively cohort of 18 healthy male participants. They then use a novel mass-spectrometry based method in muscle biopsies to quantify changes in protein levels after 14 days of unilateral limb immobilization. They highlight significantly enriched and depleted proteins and assign biological significance to their findings by running Gene Set Enrichment Analyses (GSEA).

This manuscript is exceptionally well-written, and is clear and concise. The materials and methods section is very well detailed and allows for replication of the study and data analysis. While I do think that the work is of high quality, there are some concerns that I believe should preclude publishing the article in its current form.

Author response: 

We appreciate your overall endorsement of our work. We have addressed each of your concerns point by point below, and we hope that our responses meet with your approval. 

No changes have been made to the manuscript. 

Major concerns

1. The authors provide ample data on the results of their differential (limma) analysis for protein abundance but do not supply the readers/reviewers with the raw data on which they ran the analysis (patient-level protein abundance). I believe that this could be a very valuable resource and should be provided as a supplementary data sheet.

Author response: 

Thank you for your comment. We are committed to providing open access to our raw data files, and do not wish to make any changes to the Data Availability statement provided. We have uploaded all raw data to the PRoteomics IDEntification (PRIDE) database. The uploaded data includes all:

• IDA files for library generation (*.wiff and *.wiff.scan)

• library files from IDA searches (*.txt)

• SWATH data files (*.wiff and *.wiff.scan)

 Data is available under Project accession: PXD034908 (this will be available/released upon publication according to PRIDE).

The following text has been ADDED to line 337-339 of the track changed manuscript: 

“The mass spectrometry proteomics data have been deposited to the ProteomeXchange Consortium via the PRIDE (Perez-Riverol et al., 2022) partner repository with the dataset identifier PXD034908.”

2. The authors claim that all of the participants were subject to similar dietary and training schedules in the pre-immobilization period. However, they do not mention any baseline characteristics for the participants. I think such information is also valuable to include and could give the readers a sense of the homogeneity among the cohort.

Author response: 

Thank you for this suggestion. Please note, baseline participant characteristics for height (cm), body mass (kg) and VO2peak (mL/min) were provided within the manuscript. However, in response to this comment, we have provided a table with measured and reported participant characteristics at baseline. As a result, we have put the measured information from line 121 into this new Table 1. In addition, we have provided the mean reported training frequencies and volumes for aerobic and resistance training, in Table 1.

Unfortunately, no data on habitual dietary intake were recorded in this study, given the comprehensive dietary provision prior to, and during the immobilisation period. 

The following text has been DELETED to line 121 of the track changed manuscript: 

“(25.4 ± 5.5 y, 179.4 ± 5.2 cm, 81.2 ± 11.6 kg, VO2peak: 3448.8 ± 684.4 mL/min)”

The following text has been ADDED to line 123-124 of the track changed manuscript: 

“Measured and reported participant characteristics at baseline can be found in Table 1.”

The following table has been ADDED to line 136 of the track changed manuscript: 

Table 1: Participant characteristics at baseline (mean ± SD) prior to commencing the study.

Age (y) 25.4 ± 5.5

Height (cm) 179.4 ± 5.2

Body mass (kg) 81.2 ± 11.6

VO2peak (mL∙min-1) 3448.8 ± 684.4

Peak aerobic power output (W) 242.2 ± 49.8

3RM unilateral leg press (kg) 118.1 ± 23.5 (CON)

 116.4 ± 23.6 (IMM)

Aerobic training frequency (sessions∙week-1) 1.6 ± 1.7 

Aerobic training volume (min∙session-1) 46.0 ± 58.4

Resistance training frequency (sessions∙week-1) 2.4 ± 2.0

Resistance training volume (min∙session-1) 41.8 ± 34.5

CON = control limb; IMM = immobilized limb; VO2peak = peak oxygen uptake; 3RM = three repetition maximum.

The following text has been ADDED to line 177-179 of the manuscript: 

“Habitual dietary intake of participants was not recorded in this study, given the comprehensive dietary provision prior to, and during, the immobilisation period” 

3. There are major limitations to the study design that are understandable but not well-acknowledged in the limitations section of the discussion. a) The recruitment "via electronically distributed and physical flyers between October 2016 and May 2017" could introduce a bias when it comes to the physical fitness and baseline characteristics of the participants. b) A sensible control experiment would have been to run a similar analysis on the non-immobilized leg to compare and contrast. This should be at least stated in the discussion.

Author response: 

a) As is quite typical, our study was advertised electronically to utilise the power of social media to recruit participants in our geographical area. However, as also stated, physical flyers were distributed on our university campus, including on electronic noticeboards, and in the gymnasium. We contend that by using a variety of advertising methods, we are reducing bias towards one ‘type’ of participant, which would be much more likely if this study were to be solely advertised within a gymnasium, for example. Accordingly, given a reasonable heterogeneity within the study sub-population, and that we have added additional data on participant training history, we do not believe this is a limitation.

No changes have been made to the manuscript. 

b) We certainly agree that an optimal control experiment would have been to run a similar proteomics analysis on the non-immobilized leg. However, in the interests of reducing the burden on participants (which also included two weeks (24 h/d) in a knee brace), by reducing the total number of muscle biopsies taken, our study did not utilise within-person control. Nonetheless, we have highlighted this within the limitations paragraph of our discussion. 

The following text has been ADDED to line 596-600 of the track changed manuscript: 

“In this regard, and to reduce the number of samples obtained from participants, our study did not employ a within-person control (i.e., proteomic comparison to non-immobilized leg). Therefore, our data cannot compare differences between immobilized and non-immobilized limbs at specific time points.”

4. The authors have published on a similar (but separate) topic in the following work "Effect of short-term hindlimb immobilization on skeletal muscle atrophy and the transcriptome in a low compared with high responder to endurance training model". Since transcriptomic data is available on hindlimb immobilized muscles, could the authors compare the findings on the protein level to those on the RNA level? No formal analysis is necessary, but the findings should be discussed.

Author response: 

Thank you for this comment. As requested, we have added a brief section to this manuscript on our prior findings in this selectively bred animal model, and how this may relate to findings in the present manuscript. However, this section is brief given our prior work examined dichotomous differences due to hereditary factors, that are not entirely compatible to this work. Further, the existing discussion on lines 544-550 outlines very recent and relevant data, examining the muscle transcriptome response to 4-days of knee brace immobilization in human men. 

The following text has been ADDED to line 559-567 of the track changed manuscript: 

“Indeed, the magnitude of proteolytic contribution to muscle atrophy is likely individual due to hereditary factors, and potentially training history. Our prior work in rodents selectively bred to be high or low responders to endurance training (Thompson et al., 2022), shows that superior endurance training adaptation attenuated upregulation in ubiquitination biological processes and atrophy in plantaris muscle, compared to rodents who responded poorly to endurance training. In contrast to plantaris muscle, we found no increased ubiquitination biological processes in the predominantly type I fibre soleus muscle, indicating fibre type and exercise-induced muscle phenotype, may affect the cellular response to immobilization.”

Minor concerns/comments

5. Why was the vastus lateralis muscle used for muscle biopsy? Is there a specific rationale or is it just for ease of biopsy?

Author response: 

The vastus lateralis muscle is commonly used in human research examining the effects of muscle immobilisation or disuse. Indeed, the first work to utilise any form of proteomics in human skeletal muscle (as discussed in our manuscript) examined vastus lateralis muscle (Brocca et al., 2012; Brocca et al., 2015). Furthermore, the most recent work examining transcriptomic response to immobilization also examined vastus lateralis muscle (Willis et al., 2021), as does some of the earliest work using micro-array technology (Abadi et al., 2009). As such, and to compare to these data, the vastus lateralis muscle was biopsied in the present study. 

The following text has been ADDED to line 156 of the track changed manuscript: 

“as per prior proteome analyses (Brocca et al., 2012; Brocca et al., 2015),”

6. Line 91: replace "preludes" with "precludes"

Author response: 

Thank you for identifying this error. This has been amended. 

The following text has been CHANGED on line 91 of the track changed manuscript: 

“preludes" has been replaced with "precludes".

7. Line 303: The authors mention that "A paired t-test or Wilcoxon matched pairs test" were used. Could the authors be a little more specific? If both were done, that it should be stated as such. If one was picked over the other according to the analysis, this should also be clarified.

Author response: 

Thank you for this comment. In full, our manuscript states on (now deleted) line 315: “A paired t-test or Wilcoxon matched pairs test (rectus femoris) was used for pre- to post-immobilization (MRI) comparisons…”. We appreciate you highlighting that this description was not clear enough, and we have reworded. 

The following text has been CHANGED on line 313-315 of the track changed manuscript: 

“A paired t-test (for quadriceps femoris and vastus group) and Wilcoxon matched pairs test (for rectus femoris due to non-normal distribution) was used for pre- to post-immobilization (MRI) comparisons…”

References

Abadi, A., Glover, E. I., Isfort, R. J., Raha, S., Safdar, A., Yasuda, N., Kaczor, J. J., Melov, S., Hubbard, A., Qu, X., Phillips, S. M., & Tarnopolsky, M. (2009, Aug 05). Limb immobilization induces a coordinate down-regulation of mitochondrial and other metabolic pathways in men and women. PLoS ONE, 4(8), e6518. https://doi.org/10.1371/journal.pone.0006518

Brocca, L., Cannavino, J., Coletto, L., Biolo, G., Sandri, M., Bottinelli, R., & Pellegrino, M. A. (2012, Oct 15). The time course of the adaptations of human muscle proteome to bed rest and the underlying mechanisms. Journal of Physiology, 590(20), 5211-5230. https://doi.org/10.1113/jphysiol.2012.240267

Brocca, L., Longa, E., Cannavino, J., Seynnes, O., de Vito, G., McPhee, J., Narici, M., Pellegrino, M. A., & Bottinelli, R. (2015, Dec 15). Human skeletal muscle fibre contractile properties and proteomic profile: adaptations to 3 weeks of unilateral lower limb suspension and active recovery. Journal of Physiology, 593(24), 5361-5385. https://doi.org/10.1113/JP271188

Kemp, P. R., Paul, R., Hinken, A. C., Neil, D., Russell, A., & Griffiths, M. J. (2020). Metabolic profiling shows pre-existing mitochondrial dysfunction contributes to muscle loss in a model of ICU-acquired weakness. J Cachexia Sarcopenia Muscle, 11(5), 1321-1335. https://doi.org/https://doi.org/10.1002/jcsm.12597

Perez-Riverol, Y., Bai, J., Bandla, C., García-Seisdedos, D., Hewapathirana, S., Kamatchinathan, S., Kundu, D. J., Prakash, A., Frericks-Zipper, A., Eisenacher, M., Walzer, M., Wang, S., Brazma, A., & Vizcaíno, J. A. (2022, Jan 7). The PRIDE database resources in 2022: a hub for mass spectrometry-based proteomics evidences. Nucleic Acids Research, 50(D1), D543-d552. https://doi.org/10.1093/nar/gkab1038

Thompson, J.-L. M., West, D. W. D., Doering, T. M., Budiono, B. P., Lessard, S. J., Koch, L. G., Britton, S. L., Byrne, N. M., Brown, M. A., Ashton, K. J., & Coffey, V. G. (2022). Effect of short-term hindlimb immobilization on skeletal muscle atrophy and the transcriptome in a low compared with high responder to endurance training model. PLoS ONE, 17(1), e0261723. https://doi.org/10.1371/journal.pone.0261723

Willis, C. R. G., Gallagher, I. J., Wilkinson, D. J., Brook, M. S., Bass, J. J., Phillips, B. E., Smith, K., Etheridge, T., Stokes, T., McGlory, C., Gorissen, S. H. M., Szewczyk, N. J., Phillips, S. M., & Atherton, P. J. (2021, Sep). Transcriptomic links to muscle mass loss and declines in cumulative muscle protein synthesis during short-term disuse in healthy younger humans. FASEB Journal, 35(9), e21830. https://doi.org/10.1096/fj.202100276RR

---

## [Decision Letter · Decision Letter 1]

18 Aug 2022

The muscle proteome reflects changes in mitochondrial function, cellular stress and proteolysis after 14 days of unilateral lower limb immobilization in active young men

PONE-D-21-39372R1

Dear Dr. Doering,

We’re pleased to inform you that your manuscript has been judged scientifically suitable for publication and will be formally accepted for publication once it meets all outstanding technical requirements.

Kind regards,

Suman S. Thakur, Ph.D

Academic Editor

PLOS ONE

Additional Editor Comments (optional):

Reviewers' comments:

Reviewer's Responses to Questions

**Comments to the Author**

1. If the authors have adequately addressed your comments raised in a previous round of review and you feel that this manuscript is now acceptable for publication, you may indicate that here to bypass the “Comments to the Author” section, enter your conflict of interest statement in the “Confidential to Editor” section, and submit your "Accept" recommendation.

Reviewer #1: All comments have been addressed

Reviewer #2: All comments have been addressed

2. Is the manuscript technically sound, and do the data support the conclusions?

Reviewer #1: (No Response)

Reviewer #2: Yes

3. Has the statistical analysis been performed appropriately and rigorously? 

Reviewer #1: (No Response)

Reviewer #2: Yes

4. Have the authors made all data underlying the findings in their manuscript fully available?

Reviewer #1: (No Response)

Reviewer #2: Yes

5. Is the manuscript presented in an intelligible fashion and written in standard English?

Reviewer #1: (No Response)

Reviewer #2: Yes

6. Review Comments to the Author

Reviewer #1: (No Response)

Reviewer #2: The authors appropriately addressed all my comments and concerns in this revised version of the manuscript. I have no additional comments. Congratulations!

7. PLOS authors have the option to publish the peer review history of their article (what does this mean?). If published, this will include your full peer review and any attached files.

Reviewer #1: **Yes: **Jonathan Ball

Reviewer #2: No

---

## [Editor Report · Acceptance letter]

24 Aug 2022

PONE-D-21-39372R1 

The muscle proteome reflects changes in mitochondrial function, cellular stress and proteolysis after 14 days of unilateral lower limb immobilization in active young men 

Dear Dr. Doering:

I'm pleased to inform you that your manuscript has been deemed suitable for publication in PLOS ONE. Congratulations! Your manuscript is now with our production department. 

Kind regards, 

on behalf of

Dr. Suman S. Thakur 

Academic Editor

PLOS ONE